# Validation of Automatic Cochlear Measurements Using OTOPLAN^®^ Software

**DOI:** 10.3390/jpm13050805

**Published:** 2023-05-08

**Authors:** Dimitrios Paouris, Samuel Kunzo, Irina Goljerová

**Affiliations:** Clinic of Pediatric Otorhinolaryngology of the Medical Faculty, National Institute of Children’s Diseases, Comenius University, 83340 Bratislava, Slovakia

**Keywords:** cochlea, measurement, otoplan, cochlear implant, automatic, manual, CDL

## Abstract

Introduction: Electrode length selection based on case-related cochlear parameters is becoming a standard pre-operative step for cochlear implantation. The manual measurement of the parameters is often time-consuming and may lead to inconsistencies. Our work aimed to evaluate a novel, automatic measurement method. Materials and Methods: A retrospective evaluation of pre-operative HRCT images of 109 ears (56 patients) was conducted, using a development version of the OTOPLAN^®^ software. Inter-rater (intraclass) reliability and execution time were assessed for manual (surgeons R1 and R2) vs. automatic (AUTO) results. The analysis included A-Value (Diameter), B-Value (Width), H-Value (Height), and CDLOC-length (Cochlear Duct Length at Organ of Corti/Basilar membrane). Results: The measurement time was reduced from approximately 7 min ± 2 (min) (manual) to 1 min (AUTO). Cochlear parameters in mm (mean ± SD) for R1, R2 and AUTO, respectively, were A-value: 9.00 ± 0.40, 8.98 ± 0.40 and 9.16 ± 0.36; B-value: 6.81 ± 0.34, 6.71 ± 0.35 and 6.70 ± 0.40; H-value: 3.98 ± 0.25, 3.85 ± 0.25 and 3.76 ± 0.22; and the mean CDLoc-length: 35.64 ± 1.70, 35.20 ± 1.71 and 35.47 ± 1.87. AUTO CDLOC measurements were not significantly different compared to R1 and R2 (H0: Rx CDLOC = AUTO CDLOC: *p* = 0.831, *p* = 0.242, respectively), and the calculated intraclass correlation coefficient (ICC) for CDLOC was 0.9 (95% CI: 0.85, 0.932) for R1 vs. AUTO; 0.90 (95% CI: 0.85, 0.932) for R2 vs. AUTO; and 0.893 (95% CI: 0.809, 0.935) for R1 vs. R2. Conclusions: We observed excellent inter-rater reliability, a high agreement of outcomes, and reduced execution time using the AUTO method.

## 1. Introduction

The variability of human cochlea size has been studied for many years. Measurements performed on histopathological slices at the end of the 19th and throughout the 20th century gave the first insights into cochlear anatomy and its size [1,2]. For a long time, dissecting a cochlea and directly analyzing the histopathological slices was the gold standard in cochlea anatomy research. In 1989, Takagi and Sando [3] were among the first to propose a computer-aided method to measure cochlea anatomy from histopathological slices. In 1998, Ketten and Skinner [4] used clinical CT images to analyze cochlea dimensions. Nowadays, imaging techniques such as synchrotron radiation phase-contrast imaging enable the non-destructive visualization and analysis of the organ of Corti, spiral ganglion, and dendrites at the micrometric level [5]. To achieve a generally accepted standard, researchers with experience in various fields of inner ear research and representatives of various cochlear implant manufacturers participated in consensus meetings held in Dallas in March 2005 and Asilomar in August 2005 [6]. Several coordinate systems have been introduced over the years, using different landmarks in their definition. Consequently, the results differ and cannot be easily converted or compared to each other. From the point of view of frequency analysis, the organ of Corti (OC) is the essential cochlear anatomical structure. It is used as a reference point in many cochlear anatomy and function studies concerning its tonotopic organization. The tonotopic organization of the cochlea was mathematically characterized by Greenwood [7]. The exponential formula he postulated gives the characteristic frequencies of acoustic sensitivity (in hertz) as a function of the fractional length of the OC (measured from the helicotrema) [8,9]. These same frequency distributions are used today in cochlear implants to map the frequency distribution as a function of length along the bundle. Boëx et al. [10] showed that pitch perceptions are more than one (using insertion angle) or two octaves (using insertion lengths) lower than predicted by Greenwood’s frequency position. The length of the OC cannot be directly determined in most imaging and histopathological studies of the temporal bone. Therefore, an objective concept of measuring the cochlea, closely related to the OC, was needed.

According to the above-mentioned consensus panel, the cylindrical coordinate system should be used. The requirements for this system are (A) the plane of rotation and (B) the location of the z-axis, including zero positions for the z-axis and 0 degrees for the reference angle. Within the plane of rotation, a “cochlear view” was selected based on an axis perpendicular to the basal turn of the cochlea and along the modiolus axis (x-y axis) [11,12,13]. It was decided that expanding the frame into a three-dimensional (3D) cylindrical coordinate system by adding a z-axis through the center of the modiolus—with its origin in the helicotrema—would ensure the representation of all spatial information [14]. Defining the zero reference angle (Zero Reference Angle) was also important. The desired reference point should be visible in all areas and in close relation to the OC and basilar membrane. The selected option was the center of the round window. The OC does not begin in the center of the round window, but extends approximately 2.5 mm further into the cochlear base, also called the “hook” region of the cochlea.

In cochlear implantation, due to the high variability in cochlear sizes, hence electrode arrays of different lengths, a suitable and tailored implant selection can affect performance. As spiral ganglion cell bodies have been found beyond the basal turn of the human cochlea, coverage of more apical parts of the cochlea by lateral wall electrodes might be useful to stimulate as many ganglion cells as possible, including those encoding lower frequencies [15]. Buchman et al. compared a medium-length electrode array (24 mm) and a standard-length lateral wall array (31.5 mm) from MED-EL^®^, Innsbruck, Austria. They showed a trend towards better speech perception with longer electrode arrays and a greater angular insertion depth (AID) in a prospective study in 13 CI users. Statistical significance was shown in a retrospective evaluation in a larger group of 19 CI users [16]. This result complies with Canfarotta et al., who reported a significantly better speech perception in the long-term follow-up four years after implantation in subjects implanted with an array of 31.5 mm length compared to those implanted with a shorter medium array [17,18,19,20,21]. Furthermore, recent literature also demonstrates the importance of reducing the frequency-to-place mismatch by using the post-operative anatomical information from imaging in patients’ audio processor fitting [22,23].

As mentioned above, the adaptation of the recommendations generated by the committee meetings [6] helped in the unification of methods for the measurement of the cochlear duct length (CDL), mainly relying on the “cochlear view” [24] and cochlear basal turn measurements [25,26,27]. According to the “cochlear view” principle, cochlear parameters are measured as follows:-Diameter (A-value) as the distance between the center of the round window and the furthest point on the opposite side of the lateral wall;-Width (B-value) as the distance between two furthest points at the opposite sides of the basal turn lateral wall, perpendicular to the A-value line;-Height (H-value) as the distance between the lowest part of cochlear basal turn and the tip of the apex, perpendicular to the A-value and B-value lines.

Using the measured values, the resulting CDL is calculated and used as a base for calculating the frequency distribution along the cochlea [8]. Subsequently, an appropriate electrode array can be selected based on the exact anatomical conditions. As reported in the literature, the rater’s experience and the image quality impact the final measurements, affecting the inter-rater results [28].

OTOPLAN^®^ (CASCINATION AG, Bern, Switzerland, in corporation with MED-EL, Innsbruck, Austria) is the surgical planning software for otological procedures. The software has dedicated manual, semi-automatic, and automatic planning tools for application in general otology [29] and pre-operative surgical planning and reconstruction [30,31,32,33,34,35], as well as in post-operative image analysis [23,36] in hearing implant patients. For cochlear measurements specifically, the software requires a user to manually define the cochlear view and measure the cochlea’s diameter, width, and height, resulting in automatically calculated Cochlear Duct Length (CDL_OC_) at the level of the Organ of Corti/Basilar membrane. The validity and reliability of this method amongst different users have been studied and approved [34,35]. Nonetheless, the time needed for planning can be reduced for better implementation into the daily clinical routine. Therefore, a new automatic algorithm (AUTO) has been introduced. This study aimed to compare AUTO measurements to those performed by two experienced clinicians (R1 and R2) for application in the daily clinical routine in terms of time and results’ reliability.

## 2. Materials and Methods

### 2.1. Subjects

Pre-operative images of 109 ears (56 patients) were retrospectively evaluated. The mean age was 7.3 (±3.7) years, with 53 right and 56 left ears included. The selected ears were free of any inner ear malformation.

### 2.2. Imaging

Temporal bone images obtained with the High Resolution Computer Tomography method were used in the study, having a mean (±SD) image pixel size of 0.2 (±0.1) × 0.2 (±0.1) mm^2^ with a slice thickness of 0.3 (±0.1) mm.

### 2.3. The Workflow

During the first step the patients’ CT images were loaded into the software database. Following the upload, cochlear parameters were first measured manually by two experienced otosurgeons (R1 and R2) with equal experience with the software, and compared to the automatically (AUTO) calculated parameters. For image analysis, the development version of OTOPLAN^®^ software was used [OTOPLAN^®^ v3 (1.5.0)], featuring an automated algorithm for defining the cochlea view and measuring cochlear parameters.

#### 2.3.1. Manual Method

Both raters performed the following steps in the software for each sample:Manually select the ear side;Manually rotate the imaging planes to obtain the cochlear view;Manually measure A, B and H values;The resulting CDL_OC_ is automatically calculated.

#### 2.3.2. AUTO Method

The automatic method was applied once for each sample, with the following steps:The software automatically:Detects the ear side;Reconstructs the entire inner ear, including cochlea, semi-circular canals, internal auditory canal, round window membrane and bony overhang;Sets the cochlear view;Measures A, B and H values and calculates the CDL_OC_.

The resulting screen in the software after running the AUTO method is shown in Figure 1.

### 2.4. Statistical Analysis

IBM SPSS Statistics software (IBM; Armonk, NY, USA) was used for statistical analyses. Cochlear parameters measured with R1, R2 and AUTO were tested for significance using a Wilcoxon signed-rank test, with a significance level of *p* = 0.05. The inter-rater reliability (Intraclass Correlation Coefficient—ICC was calculated) between the groups was computed using Pearson’s correlation.

## 3. Results

The time needed to perform the measurements was reduced from approximately 7 min (+/−2 min) with the manual method to less than 1 min using the automated method (*p* < 0.05).

The resulting measured values for A, B, H and CDL_OC_ are presented as mean (±SD) in Table 1 and shown in Figure 2, Figure 3, Figure 4 and Figure 5. Calculated CDL_OC_ based on the measurements obtained using the AUTO method were not significantly different from the manual measurements of R1 and R2 (Table 2) (*p* = 0.831 and *p* = 0.242, respectively), with a hypothesis of H0: Rx CDL_OC_ = AUTO CDL_OC_. The calculated intraclass correlation coefficient (ICC) for CDL_OC_ was 0.9 (95%CI: 0.85, 0.932) for R1 vs. AUTO; 0.90 (95%CI: 0.85, 0.932) for R2 vs. AUTO; and 0.893 (95%CI: 0.809, 0.935) for R1 vs. R2, which translates into a high agreement of outcomes and excellent reliability [37].

## 4. Discussion

This work aimed to compare AUTO measurements to those performed by two experienced clinicians for application in a daily clinical routine. The obtained results demonstrated no significant difference between the two approaches, as well as high interrater reliability and reduced planning time for the AUTO method.

Recent studies have shown that manual cochlear measurements performed by different experienced users may vary slightly but still produce reliable results [34,35] using the OTOPLAN^®^ software. In a study performed by Rivas et al. [38], by using a viewing patient image documentation software on a Personal Computer system and by comparing manual and automatic measurements, the results were as follows: There were significant differences between the automatic CDL measure and the calculated CDLs utilizing the manual A-value measurements. The slice containing all three anatomic landmarks was unavailable within conventional coronal, axial, or sagittal views, thus requiring specific reformatting. Approximately 90 s per ear was needed to identify the appropriate angle and measure the A-value. Finally, it was estimated by the authors that an even longer process would be necessary for clinical practice where less optimized CT analysis programs are available, and familiarity with the software is rare. In our study, by using the OTOPLAN^®^ software for manual measurements, none of the above-mentioned problems were encountered. For the manual measurement, once the “cochlear view” has been determined by the user, the program calculates and provides all the necessary values A, B, H, and CDL. Manual reformatting, as described in the work of Rivas et al., correct evaluation, and navigation through the complex inner ear anatomy, especially when performed by inexperienced users, might lead to incorrect measurements. This can negatively affect the input (A, B, H values) based on which the CDLOC value is calculated by the software, resulting in incorrect results and false predictions about the size of the cochlea and the appropriate length of the electrode array, respectively. The AUTO version overcomes all of the above-mentioned issues, including the setting of the “cochlear view”, since it is done automatically by the software.

Nonetheless, the measurements may still require a 5 to 10 min execution time depending on the experience of the user [34,35,38]. In a clinical environment, due to the overall workload, this can be a limiting factor leading to non-use of the planning tool. In the case of the inexperienced software user, the time required for measurement and possible doubts about the quality of produced results might present limitations for adopting this clinical software. With the new automatic measurement feature (AUTO), the process of cochlear parameters measurement is much faster: approximately 1 min is required for the software to perform the complete set of measurements (*A*, *B*, *H*, CDL values) and provide the results. Furthermore, independent of the user’s experience, the study showed that consistent, highly reliable measurements could be produced, compared to the manual measurements of two otologic surgeons. This increases the overall quality of the generated results and hopefully enables users with less radio-otological knowledge to embrace the software for application in their daily clinical routine.

In the present study, the inter-rater reliability of the CDL based between R1 and R2 was very high, with an ICC of 0.9. This is higher than that reported by Cooperman et al., who described inter-rater reliability with an ICC of 0.54 [39]. Furthermore, two more studies reported an even better inter-rater reliability, with an ICC value almost identical to the one in our study [34,40].

Our cochlear parameter results are comparable to the literature when measuring using version 3.0 of OTOPLAN^®^ or the “cochlear view” principle on manual measurements. In both cases, as well as when calculating the CDL value (either with OTOPLAN^®^ or manually), it is necessary to use the Alexiadis equation. Since 2006, more than 26 papers have been published on measuring the A-value (cochlear diameter), of which approximately eighteen have been published since 2019. This figure points to the increased research interest in this concept of cochlear anatomy. Khurayzi et al. [41], in their work, performed measurements in 10 ears without malformations and reviewed the literature regarding A-value measurements. Their A-value was 9.12, whereas the A-value from this literature review (3433 ears) ranged from a minimum of 7.9 mm to a maximum of 10.2 mm with a mean value of 9.13 mm (mean values of our study: manual measurements: 8.99 mm, AUTO measurements: 9.16 mm). The same author in the second study [42] provided data on 117 ears without malformations. The B-value (cochlear width) ranged from a minimum of 4.9 mm to a maximum of 7.6 mm (our mean values: manual 6.76 mm, AUTO 6.7 mm). The height of the cochlea (H-value) varied between 2 mm and 4.4 mm (our mean values: manual 3.91 mm, AUTO 3.76 mm). The corresponding CDL_OC_s, calculated using Alexiadis’ linear equation, ranged from a minimum of 26.6 mm to a maximum of 37.4 mm (our mean values: manual 35.44 mm, AUTO 35.47 mm). According to this study, pre-curved electrode arrays would not offer tight electrode contact with the modiolus in a cochlea with an elliptically shaped basal turn, as identified by a B/A ratio of <0.75. Breitsprecher et al. compared OTOPLAN^®^ version 3.0 to a specifically designed preclinical 3D reconstruction software and to the established A-value method. The results concluded that the measurements obtained from the OTOPLAN^®^ were more accurate. The width of the cochlea (B-value) compared to the diameter had a greater impact on the determination of the CDL [40].

As stated by Weber et al. [43], the chosen imaging modality showed a statistically significant effect, with MRI measurements compared to HRCT being, on average, 0.89 mm greater for the CDL_OC_ value. However, other authors considered the difference in measurements between MRI and CT images to be clinically insignificant [42,44,45]. There is a rising concern regarding the effects of ionizing radiation, especially in pediatric patients. Various studies focused on the effect of the cumulative dosage of radiation (more than 55 mGy), which can raise the risk of diseases such as leukemia or brain tumors. One particular study [46] noticed these late consequences related to exposure to ionizing radiation in the early months of life. High-resolution CT scans are considered the gold standard in inner ear imaging. As shown by many studies previously mentioned in our article, they provide detailed organ imaging. However, even if a very low relative risk outweighs the clinical benefit of CT scans, radiation doses from CT scanners should be kept at least as low as possible and alternate procedures that do not contain ionizing radiation should be considered where appropriate. In many clinics, pediatric patients who are candidates for cochlear implantation undergo an initial MR scan, and if cochlear anatomical malformations are noted, a HRCT scan is added to the radiological workup of the patient. In this way, an effort is conducted to minimize the exposure of pediatric patients to ionizing radiation. As noted by George-Jones et al. [45], a good interobserver agreement was obtained between the two raters who measured CDL with MRI images, while it was also discovered that there was a comparable performance using MRI scans versus the gold-standard CT scans. These results prove the ability of OTOPLAN to perform reliable measurements on both MRI and CT scans. The subject of this study included automatic measurements on CT scans, though a future study could also focus on automated MR measurements.

## 5. Conclusions

The authors conclude that in the analyzed 109 ears, the automated version provides reliable measurements in a setting with a significantly reduced execution time. The newest version of the planning software allows for faster and more consistent measurements of CDL, hence an appropriate choice of electrode array. Therefore, the AUTO version can be recommended for everyday clinical practice, providing a valid solution for the thorough, three-dimensional pre-operative planning of otological surgery cases.

## Figures and Tables

**Figure 1 jpm-13-00805-f001:**
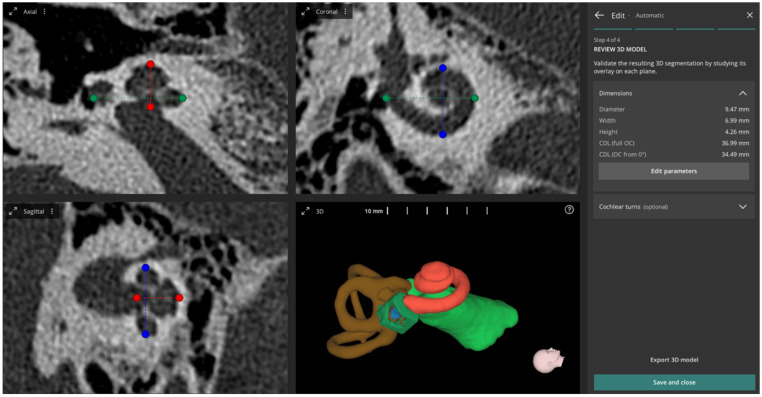
Multi-planar projection views of automatically set cochlear view including the measured diameter (A-value) (green line), width (B-value) (blue line) and height (H-value) (red line), and 3D reconstructed inner ear structures (3D view).

**Figure 2 jpm-13-00805-f002:**
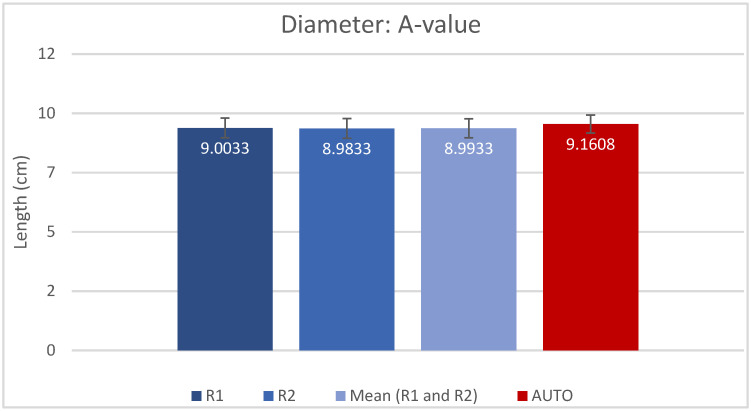
Mean A and SD, measured by rater 1 (R1), rater 2 (R2), mean of R1 and R2 and automatic algorithm (AUTO).

**Figure 3 jpm-13-00805-f003:**
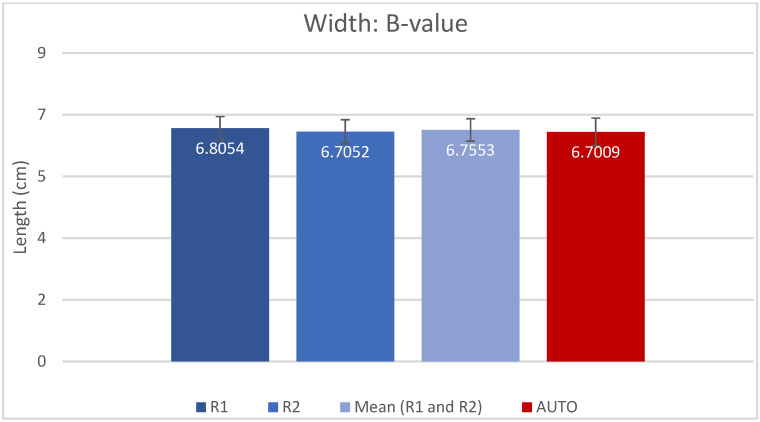
Bar chart displaying the B-value (mean and SD) measured by rater 1 (R1), rater 2 (R2), mean of R1 and R2 and automatic algorithm (AUTO).

**Figure 4 jpm-13-00805-f004:**
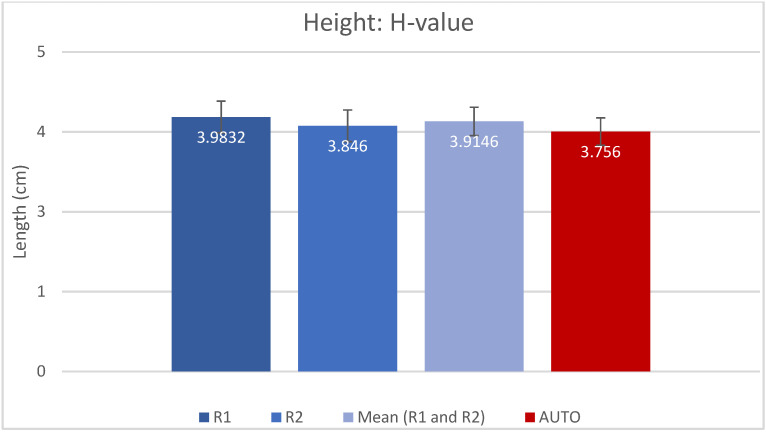
Bar chart displaying the H-value (mean and SD) measured by rater 1 (R1), rater 2 (R2), mean of R1 and R2 and automatic algorithm (AUTO).

**Figure 5 jpm-13-00805-f005:**
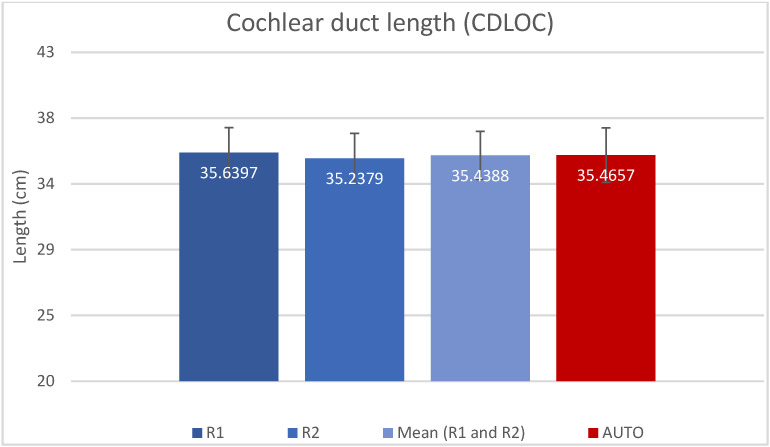
Bar chart displaying the CDL_OC_-value (mean and SD) measured by rater 1 (R1), rater 2 (R2), mean of R1 and R2 and automatic algorithm (AUTO).

**Table 1 jpm-13-00805-t001:** Mean (±SD) for diameter (A), width (B), height (H) and Cochlear Duct length (CDL_OC_) measured by R1, R2 and AUTO, respectively.

	A-Value	B-Value	H-Value	CDL_OC_
R1 (Manual)	9.00 (±0.40)	6.81 (±0.34)	3.98 (±0.25)	35.64 (±1.70)
R2 (Manual)	8.98 (±0.40)	6.71 (±0.35)	3.85 (±0.25)	35.21 (±1.71)
AUTO	9.16 (±0.36)	6.70 (±0.40)	3.76 (±0.22)	35.47 (±1.87)

**Table 2 jpm-13-00805-t002:** Statistical analysis for Cochlear Duct length (CDL_OC_) measured by R1, R2 and AUTO, respectively, with H0: Rx CDL_OC_ = AUTO CDL_OC_.

Wilcoxon Signed-Rank Test	*p*-Value
AUTO CDL_OC_ vs. R1 CDL_OC_	*p* = 0.831
AUTO CDL_OC_ vs. R2 CDL_OC_	*p* = 0.242
AUTO CDL_OC_ vs. Mean R1, R2 CDL_OC_	*p* = 0.677

## Data Availability

The data presented in this study are available on request from the corresponding author.

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
