# Peer review of "Validation of Automatic Cochlear Measurements Using OTOPLAN^®^ Software"

_jpm, 2023, doi:10.3390/jpm13050805_

Round 1

Reviewer 1 Report

Thank you for your interesting paper.

I have a question what AID (line76) means?

I have found only small typos - please add space before bracket in line 11, 12, 13, 16, 21 and 22 and after a colon in line 18, 21 and 22 for better reading. Please decide (in line 32) if the name is Takagi or Takahashi, like in references

Author Response

Line 76 issue - settled 

Line 11 to 22 typ. issues - settled

Line 32 -  the names and the referred citation have been updated accordingly

Thank you very much for your comments

Reviewer 2 Report

Dear Editor,

I reviewed the article entitled Validation of automatic cochlear measurements using 2 OTOPLAN® software By Paouris et al discussing the comparison between tha manual measurement of the cochlear parameters and those performed by the new version of Otoplan.

The article is well written (with small mistakes), consistent concerning the data and it may be useful for the scientific community dealing with hearing loss and cochlear implantation

Lines 98-99: I do not understand what the authors mean. Please rephrase.

Lines 217-218: “With the new AUTO approach, the process is much 217 faster: less than 1 minute is required for the full set of measurements. As with the fully AUTO method.” There are some problems with the spacing and I do not understand the second phrase…

Minor reviews are necessary.

Author Response

Lines 98-99:  the sentence has been rephrased

Lines 217 - 218: the sentence has been rephrased

The English language has been revised and controlled throughout the whole manuscript.

Thank you very much for your comments